# A Range-Division User Relay Selection Scheme and Performance Analysis in NOMA-based Cooperative Opportunistic Multicast Systems

**Yufang Zhang [1], Xiaoxiang Wang [1,\*], Dongyu Wang [1], Qiang Zhao [2] and Yibo Zhang [1]**

1   Key Laboratory of Universal Wireless Communication, Ministry of Education, Beijing University of Posts and Telecommunication(BUPT), Beijing 100876, China; yf910309@bupt.edu.cn (Y.Z.); dy_wang@bupt.edu.cn (D.W.); yibo@bupt.edu.cn (Y.Z.)
2   State Key Laboratory of Aerospace Dynamics, Xi'an 710043, Shaanxi, China; zqaniu@163.com
\*   Correspondence: cpwang@bupt.edu.cn

**Abstract:** The original user relay (UR) selection scheme of non-orthogonal multiple access-based cooperative opportunistic multicast scheme, which realizes inter-group cooperation between two multicast groups, ignores the distribution trend of candidate UR in the cell and adopts fixed efficient relay selection range (ERSR) to select UR. It results in high UR selection ratio. Then the coverage efficiency, defined as the ratio of successfully received users to URs, is low. To tackle this problem, a range-division user relay (RDUR) selection scheme is proposed in this paper. Firstly, it divides the circular coverage range of base station into several continuous annular areas (AAs). Secondly, different ERSRs are assigned to unsuccessfully received users in different AAs. Under different ERSR assignments, the performances of UR selection ratio and coverage ratio are analyzed. Lastly, the radius set of ERSR that optimizes system coverage efficiency is used to perform UR selection. From simulation results, with different radius sets, analytical results of UR selection ratio and coverage ratio match well with their simulated ones. It is proved that ERSR allocation affects UR selection ratio and coverage ratio. With RDUR scheme, coverage efficiency increases by at least 14% and capacity efficiency has also been improved.

**Keywords:** coverage efficiency; cooperative multicast; NOMA; path loss; relay selection scheme

---

## 1. Introduction

With the advancement of high-traffic mobile broadband services, the amount of mobile data has increased sharply. It has been estimated that mobile data will grow tenfold in the next five years [1]. The upcoming 3GPP 5G standard is designed to provide several gigabits per second throughput to users [2]. Therefore, limited spectrum resources need to be utilized more efficiently. However, it is reported that 20% popular video services produce 80% data [3], which causes the frequency band to be occupied by the same service repeatedly. Spectrum resources cannot be fully utilized. Fortunately, with the multicast technology, when multiple users request the same service, they are accessed to the same time-frequency resource. Therefore, it is worthwhile to confirm that multicast technology is effective in solving this problem.

The two-stage cooperative multicast (CM) that can jointly solve the bottleneck problem (In conventional multicast, system throughput is limited by the user with the worst channel condition.) of conventional multicast and the low coverage ratio problem of opportunistic multicast scheme (OMS) has been extensively researched in recent years. In the first stage, with OMS, BS only successfully transmits data to those multicast group (MG) users who own good channel conditions (termed as successful users (SUs)). Remaining MG users with bad channel conditions are termed as unsuccessful

---

users (USUs). In the second stage, user-to-user collaboration is carried out. Through the UR selection scheme, some SUs are selected as user relays (URs) to forward their reception signals to USUs using device-to-device (D2D) technology. It is worth mentioning that when the first-stage coverage ratio is fixed, the performance improvement of CM is mainly decided by the UR selection scheme in the second stage.

Current researches on CM are mainly based on two access technologies, orthogonal frequency division multiple access (OFDMA) and non-orthogonal multiple access (NOMA). In the OFDMA-based CM (termed as CM-OFDMA) [4–9], user-to-user collaboration is carried out only in a single MG. Users in a single MG are sparsely distributed in the cell. To ensure the cooperation between users who are far apart, the small efficient transmission range (ETR) of user's device in D2D technology is not considered as a restriction to perform UR selection.

In [4], all of the SUs are selected as URs to serve USUs. When one UR is far away from USU, due to the large path loss, he/she contributes little to the reception signal-to-noise ratio (SNR) of the USU. This causes a waste of energy. A try-best UR selection scheme is proposed in [5]. Every USU selects the nearest SU as his (her) UR. Under the assumption that SU should be closer to the BS than USU, those SUs with low average SNR are closer to USUs. They are selected as URs [6]. Several URs are placed on the fixed positions in the cell. In order to minimize system average outage probability, optimal locations are found to place these URs in the proposed genie-aided CM scheme [7]. A user-autonomous relay selection scheme is proposed in [8], in which users decide whether to participate in relay transmission in the second stage. In [9], a location-aware relay selection scheme is proposed. In order to select the least URs to achieve greater system coverage ratio than the threshold, each USU selects UR from the SUs in a certain probability. However, the method that makes USU choose UR according to the probability is not given.

Above researches only take the requirement of USU into consideration. In the second-stage D2D transmission, the small ETR restricts the cooperation among users. The UR far away from the USU can not guarantee successful reception. It results in UR redundancy. This point is neglected.

An NOMA-based CM is investigated in [10] (termed as COM-NOMA), in which with NOMA technology two MG are regarded as one quasi-MG to implement OMS in the first stage. Inter-group cooperation and intra-group cooperation are realized in the second stage. Thus, SU can be selected as UR by USU in either MG. UR selection probability increases. At the same time, the efficient relay selection range (ERSR) is set to maximal value, i.e., ETR. The SU density in the vicinity of USU is not taken into consideration.

When one USU locates in the range near the BS, where the density of SU is high, the largest ERSR covers maximum number of SU. Besides, owing to the small path loss, almost all of the SUs in the ETR of USU can be selected as UR. According to the condition of UR selection, each UR ensures that USU he (she) serves receives successfully. So the UR redundancy exists in this range. It is more serious in COM-NOMA than that in CM-OFDMA. The coverage efficiency (coverage efficiency is defined as the ratio of successful users to URs) performance is deteriorated. But in the range near the edge of the cell, where the density of SU is low, there is even no SU in the vicinity of USU, the maxmal ERSR is necessary to cover as many SUs as possible. According the different ranges where the USU locates, it is resaonable to adjust the ERSR. However, in COM-NOMA, the ERSR is fixed and equal to ETR.

Besides, when UR forwards data in the manner of D2D, the cooperative users with non-cooperative behavior, such as the selfishness of user, may lead to a sharp decline in network performance [11]. However, one assumption made by all above researches is that every MG user is accommodating. USU can be served by as many URs as requirement. Therefore, from the perspective of system, under this assumption, it is essential to reduce UR redundancy. Especially for wireless sensors network, when sensors are arranged in physical environment, power supply for them is insufficient [12,13].

Aiming at reducing UR redundancy, we propose a region-division UR selection scheme (termed as RDUR) as an enhanced scheme in COM-NOMA systems. To maximize its coverage efficiency

performance, different ERSRs are allocated to different ranges where USU locates. Firstly, the circular coverage area of the cellular is divided into continuous annular areas (AAs). Different ERSRs are assigned to AAs. USUs in the same AA have the same ERSR. Secondly, the ERSR radius set of cellular which maximizes the coverage efficiency is taken to perform UR selection.

The main contributions of this paper are summarized as follows:

- Under the influence of large-scale fading, from cell edge to center, the density of SU experiences a gradual upturn. Considering this, RDUR supplies optional radius for USU's ERSR to reduce the number of unnecessary URs. Simulation results shows that the coverage efficiency increases by at least 14% in comparision to the original UR selection scheme of COM-NOMA systems. The capacity efficiency of the second-stage D2MD has been improved. RDUR can motivate that smaller ERSRs is allocated to the cell-center AAs in high probability to reduce UR redundancy and larger ERSRs are allocated to the cell-edge AAs in high probability to ensure successful receive.
- The number of UR in the ERSR of USU after the first stage of COM-NOMA follows Poisson distribution. The parameter of Poisson distribution is derived. Simulation results show that the analytic result matches well with the simulated one. It verifies that UR redundancy exists in COM-NOMA with original UR selection scheme, when the number of MG user is high (more than 100).
- Under different ERSR radius sets, by utilizing the characteristics of Poisson distribution that its parameter is equal to its mathematical expectation, the expressions of UR selection ratio is derived. In this paper, after RDUR, the coverage ratio of the second stage is also given. Through simulation, it is supported that different ERSR assignments can effect the UR selection ratio and coverage ratio. The analytic result accurately reflects simulated one.

The rest of this paper is organized as follows. We describe the two-stage COM-NOMA in Section 2. In Section 3, RDUR is proposed. Performance analysis for different ERSR radius sets is in Section 4. Numerical simulation results and analysis are given in Section 5. Finally, we conclude this paper and highlight our findings in Section 6. Notations: Throughout the paper, the notations of mathematical variables are summarized in Table 1.

**Table 1.** Notations of mathematical variables.

| | |
|---|---|
| $R$ | The coverage radius of BS |
| $MG_i$ | The MG $i$, $i = \{1, 2\}$ |
| $\gamma$ | The distance between MG user to BS |
| $C_1$ | The coverage ratio of the first stage |
| $C_2$ | The coverage ratio of the second stage |
| $M_i$ | The number of users in MG $i$ |
| $\mathcal{S}_i$ | The SU set of $MG_i$ |
| $s_i$ | The number of SUs in $\mathcal{S}_i$ |
| $\mathcal{H}_{\mathcal{S}_i}$ | The channel gain set of $\mathcal{S}_i$ |
| $SU_{i,k_i}$ | The $k_i$-th SU from $MG_i$ |
| $H_{i,k_i}$ | The channel gain of $SU_{i,k_i}$ |
| $h_{i,k_i}$ | The small scale fading of $SU_{i,k_i}$ |
| $\gamma_{i,k_i}$ | The distance from BS to $SU_{i,k_i}$ |
| $\mathcal{H}_{\mathcal{S}_i}^w$ | The worst channel gain set of $\mathcal{S}_i$ |
| $\mathcal{S}_i^w$ | The corresponding SU set of $\mathcal{H}_{\mathcal{S}_i}^w$ |
| $P$ | The transmission power of BS |
| $P_D$ | The transmission power of UR |

**Table 1.** *Cont.*

| | |
|---|---|
| $B$ | The bandwidth originally allocated to each MG in OFDMA |
| $x_i$ | The required signal of $MG_i$ |
| $\alpha_i$ | The NOMA power allocation factors for $x_i$ |
| $N_0$ | The power of additive white Gaussian noise |
| $\delta_n^2$ | The variance of additive white Gaussian noise |
| $\sigma_0$ | The SINR threshold for UR selection |
| $\gamma_{k_i,u}$ | The distance from $SU_{i,k_i}$ to USU $u$ |
| $SINR_{k_i,u}^{x_2}$ | The reception SINR of $x_2$ from $SU_{i,k_i}$ |
| $R_0$ | The radius of ETR |
| $\Gamma_{\mathcal{U}}$ | The set of distance from BS to each USU |
| $\gamma_u^j$ | The $j$-th USU in $\Gamma_{\mathcal{U}}$ |
| $\mathcal{A}$ | The annular range where USU locates |
| $DI_{ex}^{\mathcal{A}}$ | The external diameter of annular area $\mathcal{A}$ |
| $DI_{in}^{\mathcal{A}}$ | The inner diameter of annular area $\mathcal{A}$ |
| $A_n$ | The $n$-th small annular area, $1 \le n \le N$ $\bigcup\limits_{n=1}^{N} A_n = \mathcal{A}$ and $\bigcap\limits_{n=1}^{N} A_n = \varnothing$ |
| $r_n$ | The radius of ERSR allocated to $A_n$ |
| $\theta$ | The radius set for $\mathcal{A}$ |
| $\Theta$ | The available $\theta$ set |
| $\mathcal{E}$ | The optional radius set of ERSR |
| $E_t$ | The efficient transmission distance between two users' devices |
| $W_n$ | The width of $A_n$ |
| $N^\theta$ | The total number of URs when the radius set for $\mathcal{A}$ is $\theta$ |
| $N_s^\theta$ | The number of MG users who can successfully receive when the radius set for $\mathcal{A}$ is $\theta$ |
| $C_{RDUR}{}^\theta$ | The coverage efficiency when the radius set for $\mathcal{A}$ is $\theta$ |
| $C_{RDUR}$ | The coverage efficiency of RDUR system |
| $\lambda_{MG}$ | The Poisson distribution density of MG users in the cell with radius $R$ |
| $\lambda_{A_n}$ | The MG user density in annular area $A_n$ |
| $\lambda_s^{A_n}(\gamma_u)$ | The Poisson distribution density of SU in the ERSR of $u$, when $u$ is in $A_n$ |
| $M_{UR}(\gamma_u)$ | The number of URs when USU is at $\gamma_u$ |
| $M_{UR}^{A_n}{}'$ | The total number of UR in $A_n$ |
| $M_{UR}^{A_n}$ | The number of URs in $A_n$ after expurgating repeated calculations |
| $M_s^{A_n}$ | The number of SUs in $A_n$ |
| $M_u^{A_n}$ | The number of USUs in $A_n$ |
| $R^{A_n}$ | The average repetition rate |
| $P_s(\gamma_u)$ | The probability that MG user successfully receives when the user is at $\gamma_u$ |
| $P^{A_n}$ | The probability that MG user locates in $A_n$ |
| $R_{URDR}$ | The UR selection ratio of RDUR system |
| $P_{UR_1}^{A_n}$ | The probability that USU can be served by one UR |
| $P_{UR_1}$ | The sum probability that USU is served by one UR in $\mathcal{A}$ |
| $P_{u \to s}$ | The probability that USU can successfully receive after RDUR scheme |

## 2. System Model

We consider a COM-NOMA system, which is shown in Figure 1, constituted by a BS with coverage radius $R$ and two MGs $MG_i$, $i = \{1, 2\}$. In each MG, $M_i$ users are uniformly distributed in the cellular. The probability density function of user's distance from the BS is $f(\gamma) = \frac{2\gamma}{R^2}$, $0 < \gamma \le R$. Time slot $T$ is divided into two equal sub-slots to conduct data transmission.

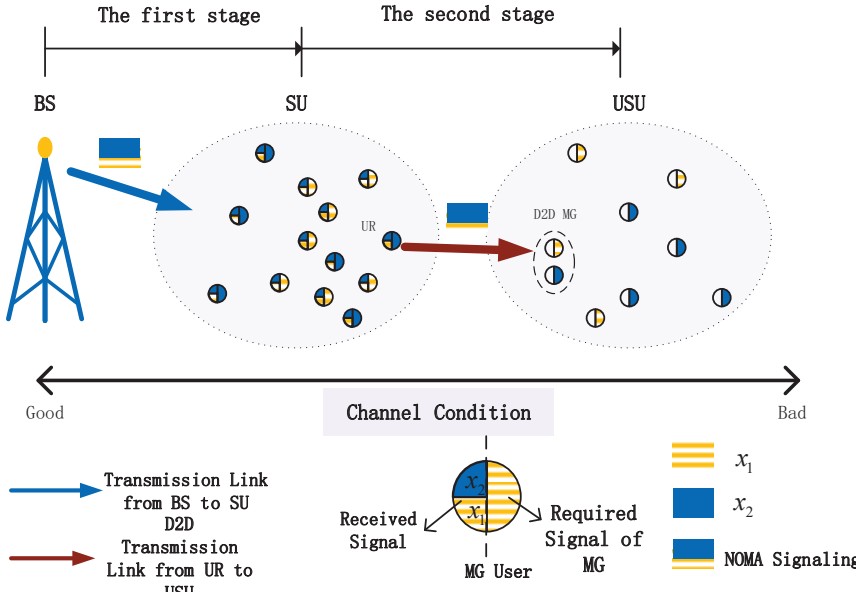

**Figure 1.** System model of cooperative multicast (COM)-based non-orthogonal multiple access (NOMA). One circle represents a multicast group user. The right side area represents the required signal. When the required signal is $x_1$ or $x_2$, this user belongs to $MG_1$ or $MG_2$ separately. The left side area represents user's reception signal. When the left side area is filled with NOMA signaling, this user is a successful user (SU). Otherwise, the user is an unsuccessful user (USU). From the perspective of received NOMA signaling, two MGs are regarded one quasi-multicast group (MG).

In order to reduce the burden of networks on keeping track of the data, especially when the number of user is large, the rateless code, such as the fountain code [14], is employed throughout the content delivery process [15].

In the first stage, two MGs are regarded as one quasi-MG. BS multicasts NOMA superposition signaling to the quasi-MG at coverage ratio $C_1$. The SU set of $MG_i$ is denoted by $\mathcal{S}_i = \{SU_{i,k_i} | 1 \leq k_i \leq s_i\}$, where $SU_{i,k_i}$ denotes the $k_i$-th SU from $MG_i$, and $s_1 + s_2 = C_1(M_1 + M_2)$. The corresponding channel gain set is $\mathcal{H}_{\mathcal{S}_i} = \{H_{i,k_i} | 1 \leq k_i \leq s_i\}$, where $H_{i,k_i} = |h_{i,k_i}|^2 \gamma_{i,k_i}^{-\beta}$. Elements in $\mathcal{H}_{\mathcal{S}_i}$ are ranged in descending order. $h_{i,k_i}$ represents small scale channel fading. $\gamma_{i,k_i}^{-\beta}$ is path loss. $\gamma_{i,k_i}$ is the distance from BS to $SU_{i,k_i}$. $\beta$ is path loss parameter. The smallest channel gain of $\mathcal{H}_{\mathcal{S}_i}$ is $H_{i,s_i}$. Without loss of generality, $H_{1,s_1} > H_{2,s_2}$ is assumed. $\mathcal{H}_{\mathcal{S}_2}^w = \{H_{2,k_2} | H_{2,s_2} \leq H_{2,k_2} < H_{1,s_1}\}$ represents the worst channel gain subset of $\mathcal{H}_{\mathcal{S}_2}$. The corresponding user set is denoted by $\mathcal{S}_2^w$. Therefore, $\mathcal{H}_{\mathcal{S}_2}^w < \mathcal{H}_{\mathcal{S}_1}$. $\mathcal{S}_1$ and $\mathcal{S}_2^w$ constitute a subgroup pair.

The required signals of two subgroups are superimposed and transmitted on the system frequency band by NOMA technology. The transmission signal of BS in the first sub-slot is

$$X_{T_1} = \sqrt{\alpha_1 P} x_1 + \sqrt{\alpha_2 P} x_2, \tag{1}$$

where $P$ is the transmission power of BS. $x_i$ is the required signal of $MG_i$. $\alpha_1$ and $\alpha_2$ are the power allocation factors (PAFs) of the paired subgroups, $\alpha_1 + \alpha_2 = 1$. Because $\mathcal{H}_{\mathcal{S}_2}^w < \mathcal{H}_{\mathcal{S}_1}$, $0 < \alpha_1 < \alpha_2 < 1$.

In OMS, in order to guarantee successful reception, for two required signals, the modulation and coding schemes (MCSs) of BS are both selected according to the least channel gain of subgroup pair, i.e., $H_{2,s_2}$. In the receiving end, after the signal allocated with large PAF is decoded successfully, the required signal can be achieved directly (corresponding to $x_2$) or after successive interference

cancellation (SIC) (corresponding to $x_1$). Therefore, the signal-and-interference-plus-noise ratio (SINR) threshold is

$$SINR_{\mathcal{S}_2} = \frac{\alpha_2 P |h_{2,s_2}|^2 \gamma_{2,s_2}^{-\beta}}{\alpha_1 P |h_{2,s_2}|^2 \gamma_{2,s_2}^{-\beta} + 2N_0}, \tag{2}$$

where $N_0$ is the power of additive white Gaussian noise (AWGN) with variance $\delta_n^2$, and $N_0 = B\delta_n^2$. $B$ is the bandwidth originally allocated to each MG in OFDMA. Because COM-NOMA occupies the bandwidth allocated to two MGs, the power of AWGN is $2N_0$.

Since users in $\mathcal{S}_2 \setminus \mathcal{S}_2^w$ own better channel conditions than those in $\mathcal{S}_2^w$, they can decode $x_2$, too. Simultaneously, it is assumed that the reception signal is abandoned by the remaining users (i.e., USUs), if only it is failed to be received. To sum up, after the first stage of COM-NOMA, each user in $\mathcal{S}_i$ obtains the NOMA signalling.

In the second stage, SU as candidate UR is selected by USU. UR conveys the NOMA signalling to USU. The transmission signal is

$$X_{T_2} = \sqrt{\alpha_1 P_1} x_1 + \sqrt{\alpha_2 P_1} x_2, \tag{3}$$

where $P_1$ is the tansmission power of UR. The MCSs are the same as the first stage.

D2D multicast (D2MD) transmission mode is introduced into the second stage. Hence, UR must locate within the ETR of USU. To make sure that the USU served by UR can successfully receive, even if USU only has one UR, SINR of signal $x_2$ provided by UR should be larger than threshold $\sigma_0$ [10], and $\sigma_0 \geq SINR_{\mathcal{S}_2}$. The UR set of USU is denoted by $\mathcal{R}_u = \{SU_{i,k_i} | \gamma_{k_i,u} \leq R_0, SINR_{k_i,u}^{x_2} \geq \sigma_0\}$. $\gamma_{k_i,u}$ is the distance from $SU_{i,k_i}$ to USU $u$. $SINR_{k_i,u}^{x_2}$ is the reception SINR of $x_2$ from $SU_{i,k_i}$. $R_0$ is the radius of ETR.

$$SINR_{k_i,u}^{x_2} = \frac{\alpha_2 P_D |h_{k_i,u}|^2 \gamma_{k_i,u}^{-\beta}}{\alpha_1 P_D |h_{k_i,u}|^2 \gamma_{k_i,u}^{-\beta} + 2N_0}. \tag{4}$$

When multiple USUs select the same one UR, they comprise a D2D MG.

## 3. The Proposed Range-Division User Relay Selection Scheme

In COM-NOMA, the density of SU is larger than the conventional CM-OFDMA scheme. Moreover, USU chooses UR from SUs who are within his/her ETR. When USU is near the cell center, the density of SU in his/her vicinity is high; otherwise, it is low. At the same time, every UR is able to provide larger SINR than $\sigma_0$ for the USU served by him (her). Therefore, near the cell center, the probability that multiple URs serve the same one USU is high, when $M_i$ is large. This is also verified by simulation in Section 5. To reduce UR redundancy and improve coverage efficiency, RDUR scheme is proposed in this section.

In RDUR, the circular coverage area of BS is divided into continuous AAs. The width of AA is decided by the optional radius of ERSR, which is no longer a fixed value $R_0$. RDUR is composed by three steps and shown in Figure 2:

(1)　Find the starting inner diameter of RDUR.
(2)　Allocate radius of ERSR to AA.
(3)　Determine the ERSR radius set.

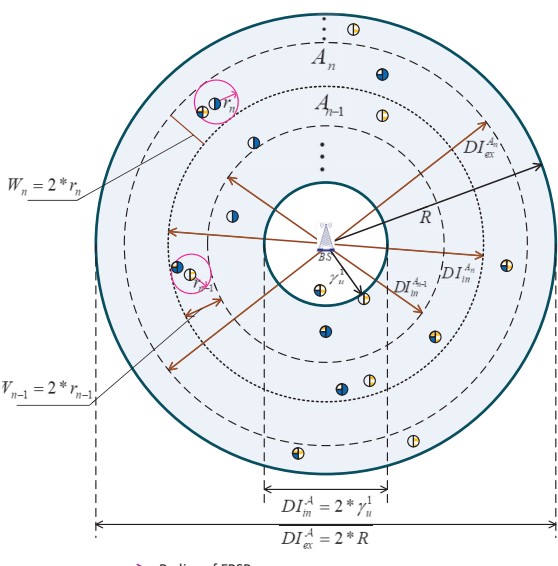

**Figure 2.** The model of range-division user relay (RDUR).

### 3.1. Starting Inner Diameter of RDUR

Since the path loss is a key factor in determining whether MG user can successfully receive in the first stage, it is important to notice that $\gamma$ exists. In the circular area, which is centered on BS and with $\gamma$ as radius, MG users are all SUs. $\Gamma_{\mathcal{U}}$ denotes the set of distance from BS to each USU, $\Gamma_{\mathcal{U}} = \left\{ \gamma_u^j | 1 \leq j \leq (1 - C_1)(M_1 + M_2) \right\}$. Without loss of generality, it is assumed that elements in $\Gamma_{\mathcal{U}}$ are ranged in ascending order.

From the analysis above, when $\gamma \leq \gamma_u^1$, there is no need to implement UR selection scheme. Hence, RDUR scheme is conducted by USUs within the annular range $\mathcal{A}$ with inner diameter $DI_{in}^{\mathcal{A}} = 2 * \gamma_u^1$ and external diameter $DI_{ex}^{\mathcal{A}} = 2 * R$. In summary, the starting inner diameter of RDUR is $2 * \gamma_u^1$.

### 3.2. Allocation of ERSR Radius

In D2D technology, USU uses global positioning system (GPS) location information and interacts with the information server (i.e., SU) [16]. ETR of a user's device is the upper bound of transmission range. Location information and training signaling of USU can be received by the SU who locates within the ETR.

For the USU who is near the cell edge, there may be no SU in the ETR. However, for the USU who is near the cell centre, several SUs may be in the vicinity. Based on this situation and the location of USU, ERSR with optional radius should be utilized to adjust the number of URs. For USU in different AAs, radius of the ERSR is different.

According to the distance between USU and BS, $\mathcal{A}$ is divided into $N$ AAs in the order from cell center to edge, $\mathcal{A} = \{A_1, A_2, \cdots, A_n, \cdots, A_N\}$, $\bigcup\limits_{n=1}^{N} A_n = \mathcal{A}$ and $\bigcap\limits_{n=1}^{N} A_n = \varnothing$. It is obvious that $DI_{in}^{A_n} = DI_{ex}^{A_{n-1}}$, where the inner diameter of $A_n$ is denoted by $DI_{in}^{A_n}$, and the external diameter is denoted by $DI_{ex}^{A_n}$. From the analysis of last Section 3.1, $DI_{in}^{A_1} = 2 * \gamma_u^1$, $DI_{ex}^{A_N} = 2 * R$.

RDUR is intended to reduce the amount of URs of cell-center USUs through narrowing the receive range of location information and training signaling. Different ERSRs are separately allocated to the USUs within $N$ AAs. The radius of ERSR allocated to $A_n$ is represented by $r_n$. The radius set for $\mathcal{A}$ is $\theta = \{r_n | n = 1, 2, \cdots, N\}$. Actually, no more than 100 $m$ distance between users' devices can be efficient in D2D technology [17]. Therefore, $r_n \in \mathcal{E}$, $\mathcal{E} = \{E_t | E_t \leq 100 \text{ m}\}$ (In this paper, $r_n$ takes discrete values less than 100 m to reduce system complexity.). $\mathcal{E}$ is the optional radius set of ERSR. The available $\theta$ set is $\Theta$.

For the ERSR set $\theta$, the relationship between the inner diameter and external diameter of $A_n$ is expressed by

$$DI_{ex}^{A_n} = DI_{in}^{A_n} + 2 * W_n, \tag{5}$$

where $W_n$ is the width of $A_n$ and $W_n = 2 * r_n$.

In summary, the relationships between inner diameters and external diameters of $N$ AAs are expressed by

$$
\begin{cases}
DI_{in}^{A_1} = 2 * \gamma_u^1, \\
DI_{ex}^{A_n} = DI_{in}^{A_n} + 2W_n, \\
DI_{in}^{A_n} = DI_{ex}^{A_{n-1}}, \\
DI_{ex}^{A_N} = 2 * R, \quad when\ DI_{ex}^{A_N} \geq 2 * R, \\
N = \min \left\{ n | DI_{in}^{A_1} + 2 \sum_n W_n >= 2R \right\}
\end{cases} \tag{6}
$$

The UR selection process is shown in Algorithm 1. Its flow chat is shown in Figure 3.

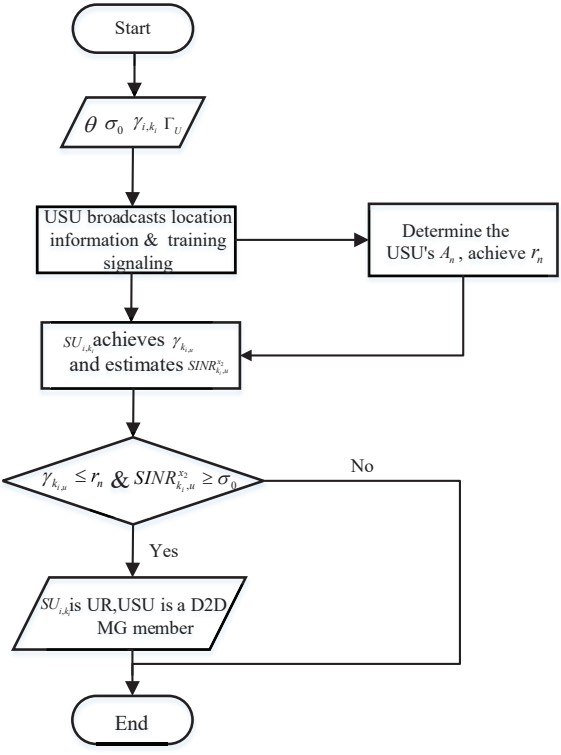

**Figure 3.** The flow chart of RDUR.

### 3.3. Determination of ERSR Radius Set

Corresponding to $\theta \in \Theta$, the total number of URs is denoted by $N^\theta$. The number of MG users who can successfully receive is $N_s^\theta$. The coverage efficiency is

$$C_{RDUR}^\theta = \frac{N_s^\theta}{N^\theta}. \tag{7}$$

In RDUR, the $\theta$, which maximize $C_{RDUR}^\theta$, is selected to perform RDUR. The system coverage efficiency given by

$$C_{RDUR} = \max_{\theta \in \Theta} \{ C_{RDRU}^\theta | \theta \}. \tag{8}$$

---

**Algorithm 1** User relay (UR) selection process.

---

1: Every MG user achieves location information.

2: Every USU broadcasts location information and a training signaling.

3: $SU_{i,k_i}$ receives location information the training signaling. The distances from $SU_{i,k_i}$ to BS and USU $u$ are measured. They are denoted by $\gamma_{i,k_i}$ and $\gamma_{k_i,u}$. Simultaneously, the channel state information of USU $|h_{k_i,u}|^2 \gamma_{k_i,u}^{-\beta}$ is also obtained by $SU_{i,m_i}$.

4: $SU_{i,k_i}$ detects the AA he (she) locates in (e.g., $A_n$) and estimates the reception SINR of each USU. If USU $u$ can satisfy $\gamma_{k_i,u} \leq r_n$ and the reception SINR of $x_2$ $SINR_{k_i,u}^{x_2}$ is greater than $\sigma_0$, this SU is selected as UR and the USU is a D2D MG member.

5: Every selected UR forwards reception signal in D2MD.

---

## 4. Performance Analysis for Different ERSR Radius Sets

Considering that not all SUs are willing to help USUs, from the perspective of system, the proposed RDUR scheme aims to select as few URs as possible. This is achieved by different ERSR allocations to AAs. For different ERSR radius sets, the performances of UR selection ratio and the second-stage coverage ratio are analyzed in this section.

### 4.1. User Relay Selection Ratio Analysis

The UR selection ratio is defined as the ratio of URs to the total users in MG. It represents the probability that a MG user is selected as UR.

Firstly, we analyze the number of URs in annular area $A_n$. In the second stage of COM-NOMA, due to the restriction of ERSR, benefiting from small path loss between the transceiver ends, the candidate UR can provide higher reception SINR than $\sigma_0$ with large probability. The probability is approximately equal to 1, which has been verified in [10]. Therefore, it is reasonable to assume that the SU within the ERSR of USU is UR. The number of UR can be obtained by analyzing the number of SU within ERSR.

When MG users are uniformly distributed in the cell, after the first stage of COM-NOMA, since ERSR is far smaller than the coverage area of cell, the number of SUs in the ERSR follows Poisson distribution. According to the character of Poisson distribution that its density is equal to the value of expectation, the average number of SUs who are in the ERSR of one USU can be calculated, when the distance between this USU and BS is $\gamma_u$.

#### 4.1.1. Average Number Of UR in the ERSR of One USU

The condition that a MG user can successfully receive is that $x_2$ assigned with large PAF can be successfully decoded from $X_{T_1}$. According to the transmission process of OMS in the first stage of COM-NOMA, for $x_2$, reception SINR should be greater than $SINR_{\mathcal{S}_2}$.

When USU $u$ locates in $A_n$, the distance from USU $u$ to BS, $\gamma_u$, satisfies $\frac{DI_{in}^{A_n}}{2} < \gamma_u < \frac{DI_{ex}^{A_n}}{2}$. Only the MG user who locates within the ERSR of $u$ has the chance to be selected as URs. Since $r_n << R$, distance from this MG user to BS approximates to $\gamma_u$. Therefore, according to the analysis in [10], the probability that MG user successfully receives is expressed by

$$P_s(\gamma_u) = \exp\left(-\frac{\gamma_u^{\beta}}{\rho_0}\delta_0\right), \tag{9}$$

where $\rho_0 = \frac{P}{N_0}$ and $\delta_0 = \frac{2SINR_{\mathcal{S}_2}}{\alpha_2 - \alpha_1 SINR_{\mathcal{S}_2}}$.

Meanwhile, MG users in the cell with radius $R$ follow Poisson distribution [18–20] with a density of

$$\lambda_{MG} = \frac{M_1 + M_2}{\pi R^2}. \tag{10}$$



Depending on the nature of the Poisson distribution, $\lambda_{MG}$ also represents the average number of MG users per unit area. When the area of ERSR is equal to $\pi r_n^2$, the MG user density in this area is

$$\lambda_{A_n} = \pi r_n^2 \lambda_{MG} = \frac{(M_1 + M_2) r_n^2}{R^2}. \tag{11}$$

Therefore, the SU in the ERSR of $u$ follows the Poisson distribution, which is verified by Figure 4, with a density of

$$\lambda_s^{A_n}(\gamma_u) = \lambda_{A_n} P_s(\gamma_u) = \frac{(M_1 + M_2) r_n^2}{R^2} \exp\left(-\frac{\gamma_u^{\beta}}{\rho_0} \delta_0\right). \tag{12}$$

The density represents the expectation of Poisson distribution. So the average number of UR in the ERSR (i.e., SU in the ERSR) of $u$ is equal to $\lambda_s^{A_n}$.

Since there is not only one USU at $\gamma_u$, and then the number of USU is calculated in the next subsection.

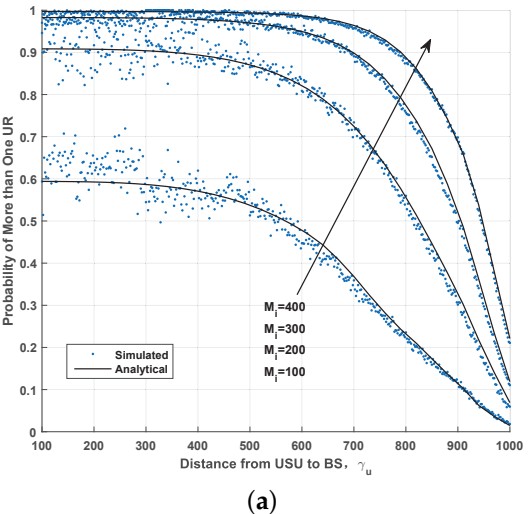

**(a)**

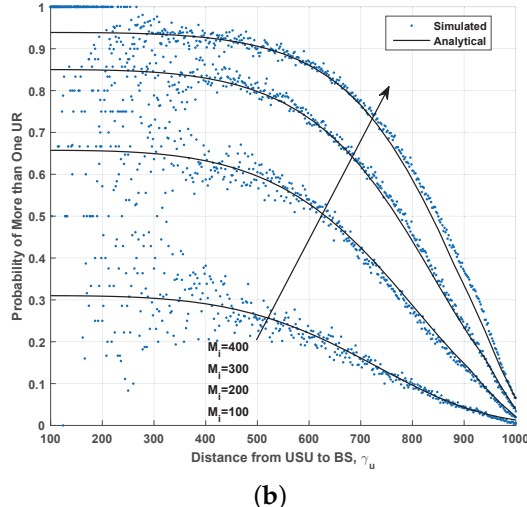

**(b)**

**Figure 4.** Probability of more than one user relay (UR) owned by USU versus the distance from him/her to base station (BS), in different number of MG users $M_i$. (**a**): $r_n = 100$ m, $\forall n \in [1, N]$. (**b**): $r_n = 75$ m, $\forall n \in [1, N]$.

### 4.1.2. Number of USU at $\gamma_u$

We analysis the number of USU at location $\gamma_u$ from BS. From (9), the probability that MG user fails to receive from BS is

$$P_u(\gamma_u) = 1 - \exp\left(-\frac{\gamma_u^\beta}{\rho_0}\delta_0\right). \tag{13}$$

At $\gamma_u$, the total number of MG user is given by

$$M(\gamma_u) = (M_1 + M_2) f(\gamma_u). \tag{14}$$

The number of USU at location $\gamma_u$ is

$$M_u(\gamma_u) = M(\gamma_u)P_u(\gamma_u) = (M_1 + M_2)\frac{2\gamma_u}{R^2}\left[1 - \exp\left(-\frac{\gamma_u}{\rho_0}\delta_0\right)\right]. \tag{15}$$

When $\frac{DI_{in}^{A_n}}{2} < \gamma_u < \frac{DI_{ex}^{A_n}}{2}$, the number of UR selected by USU in $A_n$ can be obtained by integral calculation.

### 4.1.3. Total Number of UR in $A_n$

When USU is at $\gamma_u$, the number of URs is given by

$$M_{UR}(\gamma_u) = M_u(\gamma_u)\lambda_s^{A_n}(\gamma_u). \tag{16}$$

The amount of UR in $A_n$ totals to

$$M_{UR}^{A_n}{}' = \int_{\frac{DI_{in}^{A_n}}{2}}^{\frac{DI_{ex}^{A_n}}{2}} M_{UR}(\gamma_u)d\gamma_u. \tag{17}$$

Substituting (15) and (16) into (17), the expression of the amount of UR is (18).

$$
\begin{aligned}
M_{UR}^{A_n}{}' &= \int_{\frac{DI_{in}^{A_n}}{2}}^{\frac{DI_{ex}^{A_n}}{2}} M_{UR}(\gamma_u)d\gamma_u \\
&= \int_{\frac{DI_{in}^{A_n}}{2}}^{\frac{DI_{ex}^{A_n}}{2}} (M_1 + M_2)\left[1 - \exp\left(-\frac{\gamma_u^\beta}{\rho_0}\delta_0\right)\right]\frac{2\gamma_u}{R^2}\frac{(M_1 + M_2)\,r_n}{R^2}\exp\left(-\frac{\gamma_u^\beta}{\rho_0}\delta_0\right)d\gamma_u \\
&= \frac{2r_n^2(M_1 + M_2)^2}{\beta R^4}\left[\frac{\rho_0}{\delta_0}\right]^{\frac{2}{\beta}}\left\{\left[\Gamma\left(\frac{2}{\beta}, \frac{\left(DI_{ex}^{A_n}\right)^\beta \delta_0}{2^\beta \rho_0}\right) - \Gamma\left(\frac{2}{\beta}, \frac{\left(DI_{ex}^{A_n}\right)^\beta \delta_0}{2^{\beta-1}\rho_0}\right)\right]\right. \\
&\quad \left. - \left[\Gamma\left(\frac{2}{\beta}, \frac{\left(DI_{in}^{A_n}\right)^\beta \delta_0}{2^\beta \rho_0}\right) - \Gamma\left(\frac{2}{\beta}, \frac{\left(DI_{in}^{A_n}\right)^\beta \delta_0}{2^{\beta-1}\rho_0}\right)\right]\right\}.
\end{aligned}
\tag{18}
$$

However, in the annular area, when the density of USU is higher than that of SU, the phenomenon that SU is repeatedly selected as UR exists. The number of repeatedly selected UR should be expurgated from $M_{UR}^{A_n}{}'$. We define average repetition rate (ARR) to indicate the average times that SU is repeatedly selected, which is the ratio of SUs $M_s^{A_n}$ to USUs $M_u^{A_n}$. It is denoted by

$$R^{A_n} = \frac{M_s^{A_n}}{M_u^{A_n}}, \tag{19}$$

where $M_u^{A_n}$ is

$$M_u^{A_n} = (M_1 + M_2) \underbrace{\int_{\frac{DI_{in}^{A_n}}{2}}^{\frac{DI_{ex}^{A_n}}{2}} f(\gamma_u)d\gamma_u}_{P^{A_n}} - M_s^{A_n}$$

$$= (M_1 + M_2) \frac{\left(DI_{ex}^{A_n}\right)^2 - \left(DI_{in}^{A_n}\right)^2}{4R^2} - M_s^{A_n}. \tag{20}$$

$P^{A_n}$ is the probability that an MG user locates in $A_n$. $M_s^{A_n}$ is given by

$$M_s^{A_n} = (M_1 + M_2) \underbrace{\int_{\frac{DI_{in}^{A_n}}{2}}^{\frac{DI_{ex}^{A_n}}{2}} P_s(\gamma_u)f(\gamma_u)d\gamma_u}_{P_s^{A_n}}$$

$$= (M_1 + M_2) \int_{\frac{DI_{in}^{A_n}}{2}}^{\frac{DI_{ex}^{A_n}}{2}} \exp\left(-\frac{\gamma_u^{\beta}}{\rho_0}\delta_0\right) \frac{2\gamma_u}{R^2} d\gamma_u \tag{21}$$

$$= \frac{2(M_1+M_2)}{\beta R^2}\left[\frac{\rho_0}{\delta_0}\right]^{\frac{2}{\beta}}\left[\Gamma\left(\frac{2}{\beta}, \frac{\left(DI_{ex}^{A_n}\right)^{\beta}\delta_0}{2^{\beta}\rho_0}\right) - \Gamma\left(\frac{2}{\beta}, \frac{\left(DI_{in}^{A_n}\right)^{\beta}\delta_0}{2^{\beta-1}\rho_0}\right)\right].$$

After simple algebraic operations, $R^{A_n}$ is given by (22).

$$R^{A_n} = \frac{\left[\frac{\rho_0}{\delta_0}\right]^{\frac{2}{\beta}}\left[\Gamma\left(\frac{2}{\beta}, \frac{\left(DI_{ex}^{A_n}\right)^{\beta}\delta_0}{2^{\beta}\rho_0}\right) - \Gamma\left(\frac{2}{\beta}, \frac{\left(DI_{in}^{A_n}\right)^{\beta}\delta_0}{2^{\beta-1}\rho_0}\right)\right]}{\frac{\beta}{8}\left[\left(DI_{ex}^{A_n}\right)^2 - \left(DI_{in}^{A_n}\right)^2\right] - \left[\frac{\rho_0}{\delta_0}\right]^{\frac{2}{\beta}}\left[\Gamma\left(\frac{2}{\beta}, \frac{\left(DI_{ex}^{A_n}\right)^{\beta}\delta_0}{2^{\beta}\rho_0}\right) - \Gamma\left(\frac{2}{\beta}, \frac{\left(DI_{in}^{A_n}\right)^{\beta}\delta_0}{2^{\beta-1}\rho_0}\right)\right]}. \tag{22}$$

Based on above analysis, after expurgating repeated calculations, the number of URs in $A_n$ is

$$M_{UR}^{A_n} = \begin{cases} M_{UR}^{A_n\,'}, R^{A_n} \geq 1 \\ M_{UR}^{A_n\,'} * R^{A_n}, R^{A_n} < 1. \end{cases} \tag{23}$$

After summing up the number of URs in each AA, the total number of UR is got. UR selection ratio is the ratio of the total number of UR to that of MG users and given by

$$R_{URDR} = \frac{\sum\limits_{n=1}^{N} M_{UR}^{A_n}}{M_1 + M_2}. \tag{24}$$

### 4.2. Second-Stage Coverage Ratio Analysis

As an important parameter describing multicast performance, multicast coverage ratio is defined as the ratio of total SU to MG users. Under the proposed relay selection scheme RDUR, we analyze the second-stage coverage ratio performance of COM-NOMA in this section.

From the analysis in Section 4.1, for USU, as long as there is a UR in his/her ERSR, successful reception can be guaranteed. When USU is in the annular area $A_n$ with ERSR radius $r_n$. The probability of having a UR is given by (25).

$$
\begin{aligned}
P_{UR_1}^{A_n} \\
&= \frac{1}{2\pi} \int_{\frac{DI_{in}^{A_n}}{2}}^{\frac{DI_{ex}^{A_n}}{2}} \arcsin\left(\frac{r_n}{\gamma_u}\right) \frac{4\gamma_u r_n}{R^2} \frac{2\gamma_u}{R^2} \, d\gamma_u \\
&= \frac{2r_n}{3\pi R^4} \left\{ \frac{r_n}{4} \left[ DI_{ex}^{A_n} \sqrt{\left(DI_{ex}^{A_n}\right)^2 - 4r_n^2} - DI_{in}^{A_n} \sqrt{\left(DI_{in}^{A_n}\right)^2 - 4r_n^2} \right] + \frac{\left(DI_{ex}^{A_n}\right)^3 \arcsin\frac{2r_n}{DI_{ex}^{A_n}} - \left(DI_{in}^{A_n}\right)^3 \arcsin\frac{2r_n}{DI_{in}^{A_n}}}{4} \right. \\
&\quad \left. - r_n^3 \left[ \ln\left( \frac{DI_{ex}^{A_n}}{2} \left( 1 + \sqrt{1 - \frac{4r_n^2}{\left(DI_{ex}^{A_n}\right)^2}} \right) \right) - \ln\left( \frac{DI_{in}^{A_n}}{2} \left( 1 + \sqrt{1 - \frac{4r_n^2}{\left(DI_{in}^{A_n}\right)^2}} \right) \right) \right] \right\}.
\end{aligned}
\tag{25}
$$

In $\mathcal{A}$, the sum probability that USU is served by one UR is

$$
P_{UR_1} = \sum_{n=1}^{N} P_{UR_1}^{A_n}.
\tag{26}
$$

After RDUR scheme, the probability that USU can successfully receive is given by

$$
P_{u \to s} = (1 - C_1)\left[ 1 - \left(1 - P_{UR_1}\right)^{C_1(M_1 + M_2)} \right],
\tag{27}
$$

which also is the coverage ratio of the second stage.

## 5. Simulation Results

Simulation results and analysis of the proposed RDUR scheme are shown in this section. Simulation parameters are shown in Table 2. $C_1$ influents the performance of COM-NOMA systems. When $C_1$ was less than 0.5, there was not enough SU to be selected as UR by USU, especially when USU was near the cell edge. MG users cannot fully cooperate with each other. However, when $C_1$ increased and approached to 1, the minimal channel gain of SUs who were served by BS $H_{2,s_2}$ became smaller. In multicast technology, reception rates of all SUs were limited by $H_{2,s_2}$. It was more unfair to the SU with good channel condition. Therefore, we set $C_1 = 0.6$ in the simulation. In addition, because the radius of ETR was the maxmum of $E_t$, $E_t \leq 100$ m. In order to reduce the complexity of simulation, we gave three optional radii of ERSR. Therefore, $\mathcal{E} = \{75\text{ m}, 87.5\text{ m}, 100\text{ m}\}$. To simplify simulation, it was assumed that two MGs have the same number of users, $M_1 = M_2$. This paper mainly focuses on the scenario where there is high-density MG users in the cellular. As with COM-NOMA, it was considered as high-density scenario when $M_i \geq 100$.

**Table 2.** Simulation parameters.

| Parameter Name | Value |
|---|---|
| Coverage Radius of Cellular | 1000 m |
| Radius of Efficient Transmission Range | 100 m |
| System Bandwidth | 10 MHz |
| Transmission Power of BS | 34 dBm |
| Transmission Power of User Terminal | 17 dBm |
| Noise Power Spectrum Density | $-174$ dBm/Hz |
| Path Loss Coefficient | 4 |
| Power Allocation Factor $\alpha_1$ | 0.2 |
| Power Allocation Factor $\alpha_2$ | 0.8 |
| The First-stage Coverage Ratio $C_1$ | 0.6 |
| Optional Radius Set of ERSR $\mathcal{E}$ | $\{75\text{ m}, 87.5\text{ m}, 100\text{ m}\}$ |

In the original UR selection scheme of COM-NOMA, which makes ETR as the ERSR of USU, i.e., $r_n = 100$ m, $\forall n \in [1, N]$, the probability that USU selects multiple URs is shown in Figure 4a. From the simulated results, when the distance from USU to BS was smaller than 500 m, the probability was no less than 50%. It verifies that an efficient UR selection scheme with optional ERSR to reduce UR redundancy was necessary.

In Figure 4a,b, the analytical result was calculated from the Poisson distribution of UR in the ERSR. Poisson distribution parameter is $\lambda_s^{An}(\gamma_u)$. Analytical probability that USU selects multiple URs is equal to $1 - e^{-\lambda_s^{An}(\gamma_u)} - \lambda_s^{An}(\gamma_u) e^{-\lambda_s^{An}(\gamma_u)}$. It matched well with the simulated one. Therefore, when $\mathcal{E} = \{75 \text{ m}, 87.5 \text{ m}, 100 \text{ m}\}$, it was reasonable to use the average number of UR of $u$, $\lambda_s^{An}(\gamma_u)$, to analyze the total number of UR.

In Figure 4b, when $\gamma_u$ is less than 500 m, simulated results are scattered. It was because that the number of USU near BS was small. In addition, the small ERSR leds to less URs. This results in the simulated probability value fluctuates greatly in the Monte Carlo simulation method.

From the simulated result, the closer the USU was to BS, the greater the probability was. The probability increases as $M_i$ rises. The reason is as follows. In the first stage of COM-NOMA, path loss determined whether or not MG user can succesfully receive. Therefore, the location of SU presents a trend of gathering towards BS, and the density of SU decreased as distance increased. In the second stage of COM-NOMA, the small path loss ensures that SU within USU's ERSR can provide greater reception SINR than $\sigma_0$ for USU. Therefore, the number of URs, who have been selected by an USU, is mainly decided by the density of SU. When $M_i$ rose, more SUs were located within the ERSR of USU. The probability that one USU was served by more than one UR rose.

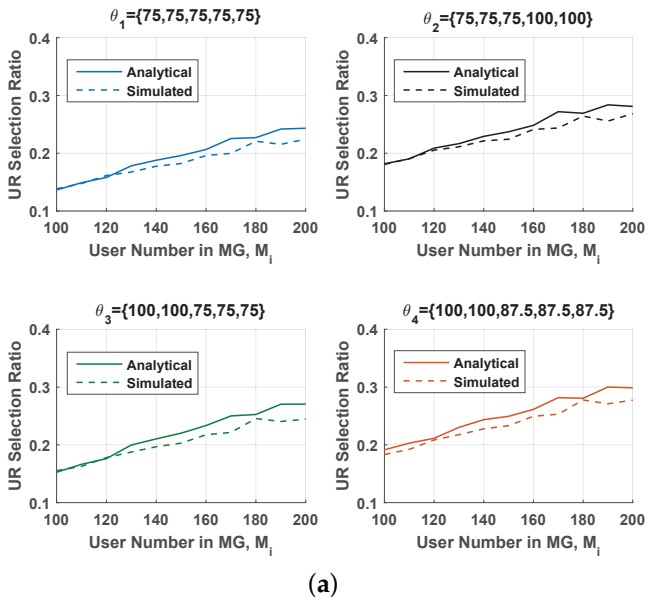

(**a**)

**Figure 5.** *Cont.*

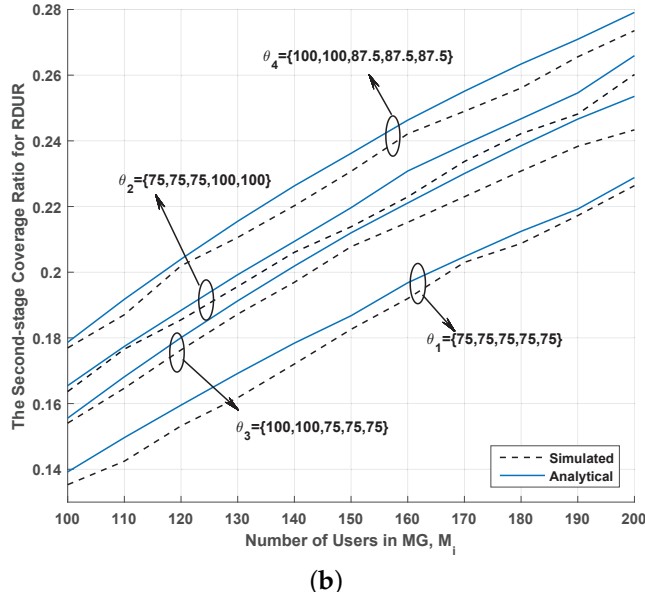

(**b**)

**Figure 5.** UR selection ratio and coverage ratio for RDUR scheme in different $\theta$. (**a**): UR selection ratio versus the number of users in MG. (**b**): The second-stage coverage ratio versus the number of users in MG.

### 5.1. Verification of Performance Analysis

In RDUR, the ERSR radius in each AA affects the number of URs. Under different ERSR radius sets, the ratio that MG user was selected as RU is plotted in Figure 5a. It shows that the analytical UR selection ratio was closed to the simulated one under different ERSR radius sets. When $M_i$ was greater than about 120, the analytical result (23) was bigger than the simulated one. Because in the AAs near the BS the number of SU was large, $R^{A_n}$ is larger than 1. Even though ARR was introduced to reduce the number of repeated UR, in these AAs some URs are repeatedly calculated. Besides, UR selection ratio increased as $M_i$ increased. This was because that the density of candidate UR in the ERSR of USU becomes large.

Obviously, as a result of small ERSR, the UR selection ratio under $\theta_1$ was smaller than that under the other three sets. It can be seen that the UR selection ratio under $\theta_2$ was larger than that under $\theta_3$. When large ERSR was allocated to those AAs close to cell edge (e.g., $r_4 = r_5 = 100$ m), owing to the fact that there were more USUs than cell-center AAs, the large ERSR contributes to select more URs than the situation that large ERSR is allocated to the cell-center AAs (e.g., $r_1 = r_2 = 100$ m). Furthermore, after a thorough comparison between $\theta_2$ and $\theta_4$, it was found that UR selection ratios were almost equal. This also verifies that different ERSR allocations can influence the number of UR.

The second-stage coverage ratio performance is presented in Figure 5b. As expected, the analytical result matches well with simulated result. The the gap between them was less than 1%, i.e., the maximum difference between the two results was less than two SUs. Furthermore, the coverage ratio under $\theta_3$ was is smaller than that under $\theta_2$. The reason was that more USU locate near the cell edge. Large ERSR was allocated to $AA_4$ and $AA_5$ to select UR. It better guarantees successful reception of USU. In addition, the second-stage coverage ratio also increased with the increase of $M_i$.

### 5.2. Performance of RDUR Scheme

In RDUR, the probability that each optional radius of ERSR $E_t$ is allocated to AA is illustrated in Figure 6 (In Figure 6, only the case when $M_i = 130$ is displayed. Those cases when $M_i$ is equal to other values were similar to that when $M_i = 130$, so they were not shown in this paper.). When the circular area covered by BS was divided into four AAs, in order from cell center to cell edge, four AAs are $A_1$,

$A_2$, $A_3$, $A_4$ and shown on the horizontal axis. Across 2000 Monte Carlo runs, it was found that a small radius of ERSR, $E_t = 75$ m, was allocated to the AA nearest to the BS, $A_1$, with the most probability 84%. A large radius of ERSR, $E_t = 100$ m, was most likely to be allocated to the cell-edge AA, $A_4$. The probability reached to 83%. It increased as the distance from BS increased. As expected, to reduce unnecessary UR, after RDUR, small ERSR should be allocated to the AA with high density of SU. To ensure that USU can select more URs and had more chances to receive successfully, large ERSR was allocated to the AA with low density of SU.

Because the proposed RDUR was an enhanced relay selection scheme aiming at NOMA-based cooperative multicast system, and the other UR selection schemes working for OFDMA-based cooperative multicast system do not make the ETR of user's device as a restriction, which results in that no matter how far the SU is from the USU, it can be selected as UR. We compared RDUR with the original UR selection scheme of COM-NOMA systems and two other UR selection schemes in Figures 7 and 8. The UR selection scheme proposed in [4] made all SUs as URs to forward data for USUs. In the try-best UR selection scheme [5], the nearest SU to each USU was selected as UR. In these two schemes, the ETR of user's device was not taken into consideration. But it indeed impacted the performance of system. Therefore, for fairness, in the simulation, the same as RDUR scheme and original scheme of COM-NOMA systems, ETR was also a restriction for these two schemes. We made the second stage as the focus of performance simulations. Because $C_1$ was a fixed value, after the first stage there are $C_1(M_1 + M_2)$ SUs. The total number of successful users was decided by UR selection scheme in the second stage. The comparison of coverage efficiency was conducted in Figure 7.

In comparison to the original scheme of COM-NOMA systems, via RDUR scheme, the coverage efficiency increased by at least about 14% under the same $C_1$. This is because USU selects URs with changeable ERSR. When the density of SU near the BS was large, the appropriate radius of ERSR was selected from $\mathcal{E}$ unlike the original UR selection scheme in COM-NOMA. In COM-NOMA, those SUs within the ETR were selected as URs. The area of relay selection was smaller in RDUR than that in COM-NOMA. Some SUs who were URs in COM-NOMA were not selected in RDUR. The number of URs reduced. In comparision to the try-best scheme, when $C_1 = 0.6$, the coverage efficiency enhances. However, when $C_1 = 0.7, 0.8$, the coverage efficiency gap between the two schemes is small. This was because when $C_1$ became larger, the distance between SU and USU decreased relatively, and the nearest SU to USU was also within his/her ERSR. Hence, via these two schemes, there was little difference in the number of URs or the number of USUs that can be successfully received. All of the SUs were selected as UEs in the scheme proposed in [4], the advantage of RDUR was obvious.

For the RDUR scheme, the coverage efficiency decreases with the increase of $C_1$. The reason is that when $M_i$ is not large enough, large $C_1$ means a small number of USU and high density of SU. The number of second-stage successful user is small and that of candidate UR is large. Correspondingly, the coverage efficiency becomes small. This also explains that why the coverage efficiency decreases with the increase of $M_i$.

To investigate the performance of RDUR further, the capacity efficiency is simulated in Figure 8. Capacity efficiency was to describe the contribution of a single UR to multicast capacity. It equalled to the ratio of the second-stage D2MD capacity (The second-stage D2MD capacity is calculated by the same method as [10].) to the total number of URs. From the simulation result, the capacity efficiency after RDUR was larger than that after the original scheme of COM-NOMA and the other two schemes. After RDUR, even though the quantity of URs was smaller than that in COM-NOMA, multicast capacity of the second stage is the sum capacity of each D2MD group, which employs UR as sender. When small ERSR was used to select URs, the distance between senders and USUs in each D2MD group decreased. In the influence of smaller loss path than original scheme, the transmission rate of D2MD is higher. The total capacity of the second stage will not decrease too much. At the same time, the number of UR was smaller than the original scheme. Thus, the capacity efficiency after RDUR scheme enhances in comparision to the original scheme. With the increase of $M_i$, the density of users

became large. Via the same UR selection scheme, more URs taking part in cooperation impaired the capacity efficiency. Therefore, capacity efficiency decreased with $M_i$.

Even though there was little difference in coverage efficiency between RDUR scheme and the try-best scheme under $C_1 = 0.7, 0.8$, the capacity efficiency performance of RDUR was better. It is because that USU can receive data from multiple URs who are within his/her ERSR. In the try-best scheme, USU only receives data from the USU nearest to him (her). The multicast capacity of the second stage after RDUR scheme is larger than that after try-best scheme.

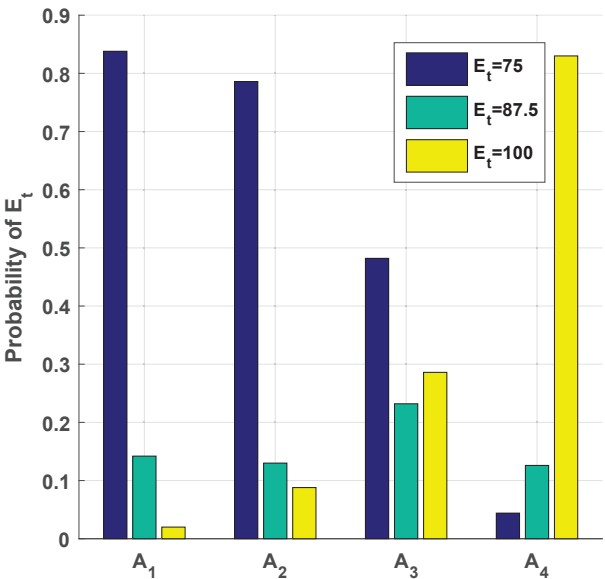

**Figure 6.** The probability of $E_t$ in different annular areas (AA), when $M_i = 130$.

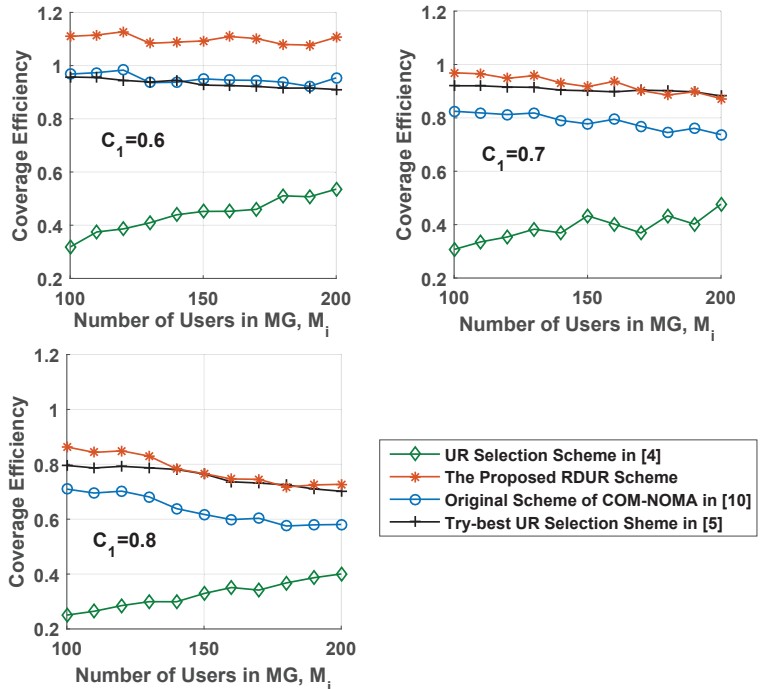

**Figure 7.** Coverage efficiency versus the number of users in MG, in different $C_1$.

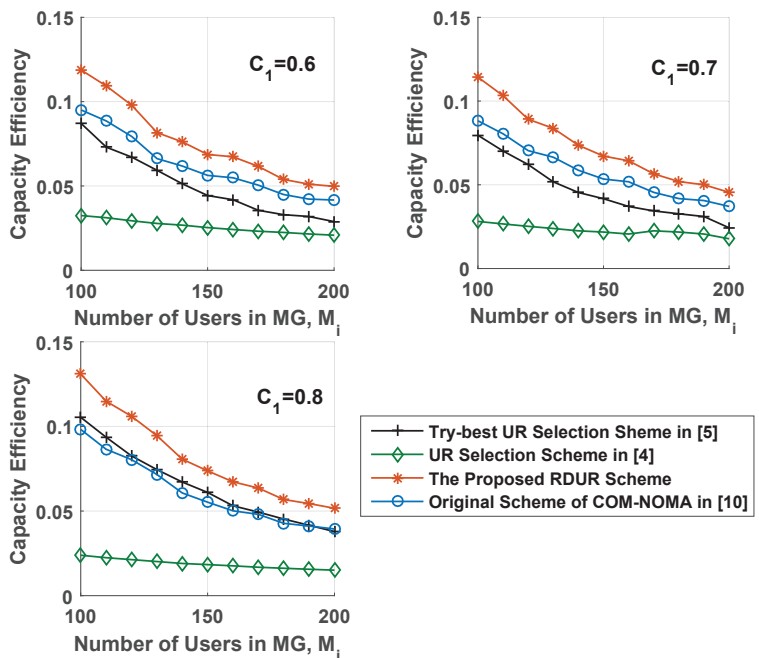

**Figure 8.** Capacity efficiency versus the number of users in MG, in different $C_1$.

## 6. Conclusions and Future Work

The two-stage COM-NOMA realizing the inter-cooperation between two MGs had improved coverage performance. However, in the second stage, some SUs were selected as URs with fixed ERSR, which is equal to the ETR of D2D. It resulted in low coverage efficiency. RDUR was proposed to solve this problem. The circular coverage range of BS was divided into several continuous AAs. Each AA was allocated corresponding ERSR. Expressions of UR selection ratio and coverage ratio are derived. The ERSR radius set that optimized system coverage efficiency was selected to perform UR selection. Main simulation results showed that for different ERSR radius sets, the analytical results of UR selection ratio and coverage ratio matched well with their simulated results. In comparision to other UR selection schemes, via RDUR, the capacity efficiency of the second-stage D2MD had been improved. The coverage efficiency was at least 14% higher than the original UR selection scheme of COM-NOMA systems.

Through RDUR, the coverage efficiency of COM-NOMA systems was improved. However, the radius of ERSR could only be chosen from several fixed values. It diminishes coverage efficiency of COM-NOMA system. In fact, in oder to maximize the coverage efficiency, USU should adaptively achieve the radius of ERSR according to his(her) own location information. In the future, a UR selection scheme with adaptive ERSR will be researched. Any distance smaller than 100 m can be chosen as the ERSR radius by USU.

**Author Contributions:** Conceptualization, Y.Z. (Yufang Zhang) and D.W.; methodology, Y.Z. (Yufang Zhang) and X.W.; software, Y.Z. (Yufang Zhang) and Q.Z.; validation, Y.Z. (Yufang Zhang) and Q.Z.; formal analysis, Y.Z. (Yufang Zhang); investigation, Y.Z. (Yufang Zhang) and Y.Z. (Yibo Zhang); writing—original draft preparation, Y.Z. (Yufang Zhang); writing—review and editing, Y.Z. (Yufang Zhang); visualization, Y.Z. (Yufang Zhang).

**Funding:** This research was funded by the National Natural Science Foundation of China under Grant number 61701038.

**Conflicts of Interest:** The authors declare no conflict of interest.

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
