# Peer review of "A Range-Division User Relay Selection Scheme and Performance Analysis in NOMA-based Cooperative Opportunistic Multicast Systems"

_electronics, doi:10.3390/electronics8050544_

Round 1

Reviewer 1 Report

The authors describe a Range-Division User Relay (RDUR) selection strategy to select the Efficient Relay Selection Range (ERSR) within which unsuccessful users (USUs) can elect user relays (URs) to receive content through D2D techniques in non-orthogonal multiple access (NOMA) cooperative multicast scenarios.

The paper introduces a mathematical model of the proposed selection scheme, which is validated through simulation using the Monte Carlo method. Simulation results show that the presented theoretical model is fairly accurate when different parameters are changed, including the number of users in the multicast group, the coverage ratio, and the selected ERSR.

The paper is well written, all necessary background is given, and it is easy to follow the authors' reasoning and the mathematical analysis.

Sometimes the English feels fractured, but it does not prevent understanding.

Given the high number of mathematical variables in the model, a table summarizing such variables and their meaning would make following the mathematical steps easier, as it would save the reader from the need to search the text for the definition or the meaning of variables.

Author Response

1.  The paper is well written, all necessary background is given, and it is easy to follow the authors' reasoning and the mathematical analysis.Sometimes the English feels fractured, but it does not prevent understanding.

Response: We thank Reviewer for pointing this out.

We have proofread the manuscript and  revised some expressions. They are marked in blue in the manuscript.  

2. Given the high number of mathematical variables in the model, a table summarizing such variables and their meaning would make following the mathematical steps easier, as it would save the reader from the need to search the text for the definition or the meaning of variables

Response: We thank Reviewer for this suggestion. It is very helpful to improve the readability of my manuscript. Therefore, we summarize the symbols  of mathematical variables and their meaning in Section Introduction.

Reviewer 2 Report

A range-division user relay selection scheme (termed as RDUR) is proposed in this paper. From simulation results, with different radius sets, analytical results of UR selection ratio 12 and coverage ratio match well with their simulated ones. It is proved that ERSR allocation affects UR 13 selection ratio and coverage ratio.  The paper is an interest but it will need to improve:

1) The novelty is not compared with existing method.

2) The application domain will need to clearly mentioned considering different papers such as:

    -Detection of primary user emulation attack in sensor networks

    -Information prediction in sensor networks using Milne-Simpson’s scheme

   --Adaptive channel estimation techniques for MIMO OFDM systems

 --Power control algorithm for cognitive radio systems

3) More simulation results considering cyber attack will be expected.

4) The limitation of the proposed method will need to mention.

5) The simulation setting will need to expand.

6) A flow chart will need to include in simulations.

7) Before describing mathematics it will need to describe  the way in text.

Author Response

1. The novelty is not compared with existing method.

Response: We thank Reviewer for pointing this out. In order to introduce the novelty better, we simulate two other existing mothords \cite{FenHou,Yiqingzhou20176} as shown in Figures.\ref{fig: CU} and Figure.\ref{fig: CaU}. These two figures and added analyses are in the section \emph{Simulation Sesults}.

2. The application domain will need to clearly mentioned considering different papers such as:\\

-Detection of primary user emulation attack in sensor networks\\

-Information prediction in sensor networks using Milne-Simpson’s scheme\\

--Adaptive channel estimation techniques for MIMO OFDM systems\\

--Power control algorithm for cognitive radio systems

Response: It is true as Reviewer suggested. We reference these papers and point out the application domain in my artical. The reduction in relay redundancy is really necessary in wireless sensor networks. So we cite article ``Detection of primary user emulation attack in sensor networks"\cite{WSN2} and ``Information prediction in sensor networks using Milne-Simpson’s scheme"\cite{WSN1}, and add the application domain in Section \emph{Introduction}.

Page 2 Line 75: Especially for wireless sensors network,  when sensors are arranged in physical environment,  power supply for them is insufficient[12,13].

3. More simulation results considering cyber attack will be expected.

 Response: We feel great thanks for your  review work on our manuscript.   As an enhanced UR selection scheme of COM-NOMA systems, the RDUR scheme is proposed in this artical. The utilization of NOMA technology makes multiple users  be able to receive the same one NOMA signaling. However, the signaling is superposed by several signals required by different users. That is to say,  each user also receives others' signals. This is really vulnerable to cyber attacks. However, this artical is aiming at reducing UR redundancy and enhancing the coverage efficiency of COM-NOMA systems. To maximize its coverage efficiency performance, different ERSRs are allocated to different AAs where the USU locates. And the simulation results also verified the efficiency of the proposed scheme. The  cyber attack will be considered into our future work as the further  research on  the COM-NOMA systems.

4.  The limitation of the proposed method will need to mention.

Response: We thank Reviewer for pointing this out. In the proposed method,  the number of optional radius of ERSR is small.  And their values are several fixed number. In fact, in oder to maximize the coverage efficiency of COM-NOMA system, USU should adaptively achieve the radius of ERSR according to his(her) own location information.

We  point this point out in Section \emph{Conclusion}.

Page 18 Line 349: \emph{However, the radius of ERSR could only be chosen from several fixed values. It deminishes coverage efficiency of COM-NOMA system.  In fact, in oder to maximize the coverage efficiency, USU should adaptively achieve the radius of ERSR according to his(her) own location information. In the future, a UR selection scheme with adaptive ERSR will be researched. And any distance smaller than $100 m$ can be chosen as the ERSR radius by USU.} }}\\

5. The simulation setting will need to expand.

Response: We thank Reviewer for this suggestion.  To expand the simulation

parameter, we add the following content  in  Section Simulation Results.

Page 12 Line 226:  $C_{1}$ influents the  performance of COM-NOMA systems.  When   $C_{1}$ is less than 0.5,  there is not enough SU  to be selected as UR by USU, especially when USU is near the cell edge.  MG users cannot fully cooperate with each other. However, when $C_{1}$ increases and approaches to 1, the minimal channel gain  of SUs who are served by BS $H_{2,s_{2}}$ becomes smaller. In multicast technology, reception rates of all SUs are limited by $H_{2,s_{2}}$.  It is more  unfair to the SU with good channel condition.  Therefore, we set $C_{1}=0.6$ in the simulation. In addition, because the radius of ETR is the maxmum of  $E_t$,  $E_t\leq100\ m$. In order to reduce the complexity of simulation, we give three optional radius of ERSR. Therefore, ${\cal E}=\{75\ m, 87.5\ m, 100\ m\}$.

6. A flow chart will need to include in simulations.

Response: It is really true as Reviewer suggested that a flow chart is needed. We draw the  flow chart Figure.\ref{fig: FC} of UR selection process for ease of simulation. It is in Page 8 of my mamuscript.

Page 7 Line 180: And its flow chat is shown in Figure.3.

7. Before describing mathematics it will need to describe  the way in text.

Response: We thank Reviewer for this suggesstion. Considering  the suggestion, we add explanation for the mathematics.

Page 9 Line 196: \emph{When MG users are uniformly distributed in the cell, after the first stage of COM-NOMA, since ERSR is far smaller than the coverage area of cell, the number of SUs in the ERSR follows Poisson distribution. According to the character of Poisson distribution that its density is equal to the value of expectation,   the average number of SUs who are in the ERSR of one USU can be calculated, when the distance between this USU and BS is ${\gamma_u}$.}

Page 10 Line 208: Since there is not only one USU at ${\gamma_u}$, and then the number of USU is calaulated in the next subsection.

Page 10 Line 211: When $\frac{{DI_{in}^{{A_n}}}}{2} < {\gamma _u} < \frac{{DI_{ex}^{{A_n}}}}{2}$,  the number of UR selected by USU in $A_n$ can be obtained by integral calculation.

Page 11 Line 215: After summing up the number of URs in each AA, the total number of UR is got.   UR selection ratio is the ratio of the total number of UR to that of MG users and given by

}}
